# Lipschitz Continuity for Harmonic Functions and Solutions of the $\bar{\alpha}$-Poisson Equation

**Miodrag Mateljević** [1,†]**, Nikola Mutavdžić** [2,*,†] **, Adel Khalfallah** [3,†],

1    Serbian Academy of Science and Arts, Kneza Mihaila 35, 11000 Belgrade, Serbia; miodrag@matf.bg.ac.rs
2    Mathematical Institute SANU, Kneza Mihaila 36, 11000 Belgrade, Serbia
3    Department of Mathematics, King Fahd University of Petroleum and Minerals, Dhahran 31261, Saudi Arabia; khelifa@kfupm.edu.sa
*    Correspondence: nikola.math@gmail.com
†    These authors contributed equally to this work.

**Abstract:** In this paper we investigate the solutions of the so-called $\bar{\alpha}$-Poisson equation in the complex plane. In particular, we will give sufficient conditions for Lipschitz continuity of such solutions. We also review some recently obtained results. As a corollary, we can restate results for harmonic and $(p,q)$-harmonic functions.

**Keywords:** Poisson's kernel; Green function; harmonic functions; gradient estimate; lipschitz continuity

**MSC:** 31A05; 30C40; 35J25

## 1. Introduction and Preliminaries

For a positive weight function $\rho$ on the unit disc $\mathbb{D}$, we define the operators $L_\rho$ and $L_\rho^*$ by

$$L_\rho = D_z(\rho D_{\bar{z}}) \quad \text{and} \quad L_\rho^* = D_{\bar{z}}(\rho D_z).$$

In [1], these operators are called the *weighted Laplacian operators.* For the weight function

$$\rho = \rho_\alpha(z) = (1 - |z|^2)^{-\alpha}, \quad z \in \mathbb{D} = \{z \in \mathbb{C} : |z| < 1\}, \quad \alpha > -1,$$

the operator $L_\rho$ is called the *standard weighted Laplacian* and is denoted by $L_\alpha$ for simplicity. Likewise, the operator $L_\rho^*$ is denoted by $\overline{L_\alpha}$.

In analogy with the Poisson equation $L_0 u = g$, the $\bar{\alpha}$-Poisson equation is defined on $\mathbb{D}$ as

$$\overline{L_\alpha} u = g, \tag{1}$$

where the function $g$ is given on the unit disc $\mathbb{D}$. Assume that $g$ is continuous on the disc $\mathbb{D}$ and that $\rho_\alpha^{-1} g$ is bounded. Our main result states that, if a solution $u$ of the $\bar{\alpha}$-Poisson Equation (1) has a continuous extension to the unit circle $\mathbb{T}$ that is Lipschitz on $\mathbb{T}$, then $u$ is Lipschitz on the entire unit disc $\mathbb{D}$. The Poisson equation is a fundamental problem in classical literature. For example, the book [2] considers elliptic partial differential equations of the second order, which are uniformly strongly elliptic. Since the operator $\overline{L_\alpha}$ is not uniformly elliptic, we can not apply these classical methods; see for example [3], where the first two authors of this paper showed that the corresponding analogue of the Hopf lemma is false.

Harmonic quasiconformal mapping (shortly HQC-for definition and properties of quasiconformal mappings in $\mathbb{R}^n$ see [4,5]) of the unit disk are related to the context of this paper, and the subject related to HQC mappings is now an active area of research; in particular, it has been intensively studied with Belgrade Analysis group, for example, [6–9] and the literature cited therein and in this paper. Particularly, paper [7] studies quasiconformal

diffeomorphisms $f : G_1 \to G_2$, (where $G_1, G_2$ are domains with $C^2$-smooth boundaries), which are also a solution of the (classical) Poisson's equation. In this case, it is proven that all partial derivatives of $f$ are bounded on $G_1$, i.e., that such a mapping is Lipschitz. For more details see Section 4. This particular result is, in some sense, a spatial version of the famous Kellogg's theorem. In the short terms, Kellogg's theorem is related to the boundary behavior of conformal mapping $f$ between two $C^{1,\beta}$, $0 < \beta < 1$ plane domains. Roughly speaking, the plane Jordan curve is $C^{1,\beta}$ smooth if its arc length parametrization has a $\beta$-Hölder first derivative. The conclusion of this theorem is that the complex derivative of $f$ has a $\beta$-Hölder extension to the boundary [10,11]. Many generalizations of this classical result were obtained by various mathematicians. In a broader sense, this topic is connected to the gradient estimates of spatial harmonic functions [12]. Deeper origins of this topic can also be seen in the famous Schwartz lemma, and some newer result and history of this area can be found in [3]. For additional results, it is important to mention [13,14], where Lipschitz continuity of the solution of the hyperbolic Poisson's equation and $(a, b)$–harmonic functions are investigated.

### 1.1. A Short Preview of This Article

First, we consider some basic properties of $\alpha$–harmonic mappings. In particular, we improve on the results of Chen and Kalaj [15]. Behm [16] found the Green function and provided a solution for the Dirichlet boundary value problem in the case of the $\alpha$–Poisson equation. Our method is based on Theorem 8, which gives an estimate of the Green potential $G_\alpha$ of $g$. At the beginning of this paper, we will introduce a basic notation together with a definition of the so-called $\alpha$–Laplacian and $\alpha$–harmonic functions. Also, the definition and properties of $\alpha$–Poisson's kernel and $\alpha$–Poisson's integral are stated, as a very important technical asset used in our research. More information about this notion can be found in Olofsson's and Wittsten's paper [1]. After that, we recall the definition of the *Green function for the $\alpha$–Laplacian*, which is thoroughly investigated in Behm [16]. A formulation and a solution for the Dirichlet boundary value problem in the case of $\alpha$–Poisson's equation are presented and proven in Chen and Kalaj's paper [17], which demonstrates Theorem 1. In paper [18], Chen used this result to prove the necessary and sufficient condition on the boundary function for Lipschitz continuity of an $\bar{\alpha}$-harmonic mapping and proved Theorem 2.

The first result of this paper is weakening the assumption on the boundary value of an $\bar{\alpha}$-harmonic mapping $v$, which is written in part $(iv)$ of Theorem 2 [15] , and obtaining Theorem 6. In fact, since $\mathcal{S}_\alpha(L^\infty(\mathbb{T})) \subset L^\infty(\mathbb{D})$ for $\alpha > 0$, by Claim 1, we proved that condition $\mathcal{S}_\alpha[f'] \in L^\infty(\mathbb{D})$ is unnecessary. The proof of Theorem 6 uses the Hardy space technique and it can be found in the first author's monography [4], and Theorem 5 is proven in first author's and A. Khalfallah's paper [19]. Also, Theorem 7 gives another form of the part $(i)$ of Theorem 2, which considers $(p, q)$–harmonic mappings, as well as Hölder continuous boundary values. The second improvement of Theorem 2 considers the condition on $g = -\overline{L_\alpha}u$. This result is proven in Theorem 8, and uses various estimates, which we establish in Section 2.3.

### 1.2. $\alpha$–Harmonic Mappings

Let $u$ be a $C^2$ function on $\mathbb{D}$. Recall that two complex derivatives $\frac{\partial}{\partial z} = D_z$ and $\frac{\partial}{\partial \bar{z}} = D_{\bar{z}}$ of $u$ are written by

$$\frac{\partial}{\partial z}u = D_z u = (u_x - iu_y)/2 \quad \text{and} \quad \frac{\partial}{\partial \bar{z}}u = D_{\bar{z}}u = (u_x + iu_y)/2$$

respectively, where $z = x + iy$.

For the weighted Laplacian defined above, we have

$$L_\rho^* u = \overline{L_\alpha}u = D_{\bar{z}}\rho D_z u + \rho D_{z\bar{z}}u = \alpha(1 - |z|^2)^{-\alpha-1}zu_z + (1 - |z|^2)^{-\alpha}u_{\bar{z}z},$$

$$L_\alpha u = \alpha(1 - |z|^2)^{-\alpha-1}\bar{z}u_{\bar{z}} + (1 - |z|^2)^{-\alpha}u_{\bar{z}z}.$$

First, we can see that $L_\alpha^* u = 0$ if

$$(1 - |z|^2)u_{\bar{z}z} + \alpha z u_z = 0. \tag{2}$$

Moreover, $L_\alpha u = 0$ if $L_\alpha^* \bar{u} = 0$. For $a, b \in \mathbb{R}$, which cannot be negative integers and which satisfies $a + b > -1$, the operator is defined in [14] as

$$L_{a,b}u = (1 - |z|^2)u_{\bar{z}z} + a z u_z + b \bar{z} u_{\bar{z}} - abu. \tag{3}$$

Let us recall the notion of $(a, b)$-harmonic functions. A function $u$ is said to be $(a, b)$-harmonic if $u \in C^2(\mathbb{D})$ and $L_{a,b}u = 0$.

It is clear that $(0, \alpha)-$harmonic functions are $\alpha$-harmonic functions, and $(\alpha, 0)-$harmonic functions are $\bar{\alpha}$-harmonic functions.

**Definition 1.** *Let $G$ be a bounded subset of $\mathbb{C}$. A function $f : G \to \mathbb{C}$ is $\beta$-Hölder continuous on $G$ where $0 < \beta < 1$ if there exists $c' > 0$ such that*

$$|f(z_1) - f(z_2)| \leqslant c'|z_1 - z_2|^\beta, \quad z_1, z_2 \in G.$$

*We say that $f$ is Lipschitz continuous on $G$ if there exists $c'' > 0$ such that*

$$|f(z_1) - f(z_2)| \leqslant c''|z_1 - z_2|, \quad z_1, z_2 \in G.$$

Set $p = u_z$ and $q = u_{\bar{z}}$. Since $\overline{u_{\bar{z}}} = \overline{u}_z$ and $\overline{u_z} = \overline{u}_{\bar{z}}$, we find $z u_z$ and $\bar{z} u_{\bar{z}}$ are conjugates of each other, and also $u_{\bar{z}z} = q_z$ and $\overline{u}_{\bar{z}z} = \overline{q}_{\bar{z}} = \overline{q_z}$; therefore, $u_{\bar{z}z}$ and $\overline{u}_{\bar{z}z}$ are conjugate. It is easy to check that $u_{\bar{z}z} = \frac{1}{4}\Delta u$, where $\Delta u = \frac{\partial^2 u}{\partial x^2} + \frac{\partial^2 u}{\partial y^2}$.

If we set $d(z) = 1 - |z|^2$, then $\rho_\alpha = d^{-\alpha}$, and by easy computation we find

$$\rho_z = \alpha d^{-\alpha}\bar{z}, \quad \rho_{\bar{z}} = \alpha d^{-\alpha-1}z, \quad \rho_x = 2\alpha d^{-\alpha-1}x \quad \text{and} \quad \rho_y = 2\alpha d^{-\alpha-1}y.$$

Since $2\rho D_z u = \rho(u_x - iu_y)$, we find

$$4L_\rho = D_x[\rho(u_x - iu_y)] + iD_y[\rho(u_x - iu_y)] = D_x(\rho u_x) + D_y(\rho u_y) + i(\rho_y u_x - \rho_x u_y).$$

Hence,

$$4L_\rho = \rho\Delta u + \rho_x u_x + \rho_y u_y + i(\rho_y u_x - \rho_x u_y).$$

If $u$ is a real-valued function, then $L_\rho u = 0$ if $\rho\Delta u + \rho_x u_x + \rho_y u_y = 0$ and $yu_x - xu_y = 0$, that is

$$\Delta u + 2\alpha\rho_1(xu_x + yu_y) \text{ and } yu_x - xu_y = 0.$$

The general solution of the equation $yu_x - xu_y = 0$ is $u = f(x^2 + y^2)$. Since $\rho u_z = \rho g(r)\bar{z}$, we find $\rho g(r)r^2 = zF(z) = c$, and hence, $F = 0$ and $u_z = 0$. Thus, $u = c$.

If a function $u \in C^2(\mathbb{D})$ satisfies the $\alpha$-harmonic equation

$$L_\alpha(u) = 0,$$

then it is said to be an $\alpha$-harmonic mapping. In the case $\alpha = 0$, $\alpha$-harmonic mappings are just Euclidean harmonic mappings. In the literature, $L_\alpha$ is sometimes denoted as $\Delta_\alpha$.

Set $u = pdz + qd\bar{z}$. We can rewrite $\Delta_\alpha u = (\rho_\alpha q)_z$ in the form $\overline{\Delta_\alpha u} = (\rho_\alpha \bar{q})_{\bar{z}}$. Hence, if $u$ is $\alpha$-harmonic, then there is a holomorphic function $f$ such that $\rho_\alpha q = \bar{f}$.

Next, by computation, we find $u_\theta = piz - iq\bar{z}$, $u_r = pe^{i\theta} + qe^{-i\theta}$ and

$$e^{i\theta}\rho_\alpha(iru_r - u_\theta) = 2ir\bar{f}, \qquad \rho_\alpha(iru_r - u_\theta) = 2i\bar{z}\bar{f}.$$

At first glance, we would like to conclude that, if $u$ is real-valued, then $\rho_\alpha u_\theta$ and $ir\rho_\alpha u_r$ are Euclidean conjugate harmonic. However, it seems that every real-valued $\alpha$-

harmonic mapping is constant. By Riesz's theorem on conjugate functions (see Rudin [20], Theorem 17.26), there exists a finite constant $A_p$ such that

$$M_p(r, ru_r) \leqslant A_p M_p(r, u_\theta).$$

*1.3. $\alpha$-Poisson's and $(p, q)$-Poisson's Integral*

Let us recall that the classical Poisson kernel and Poisson integral are given by

$$P(z) = \frac{1 - |z|^2}{|1 - z|^2} \quad \text{and} \quad \mathcal{P}[f](z) = \frac{1}{2\pi} \int_0^{2\pi} u(e^{i\theta}) \frac{1 - |z|^2}{|z - e^{i\theta}|^2} d\theta.$$

Olofsson and Wittsten showed in [1] that, if an $\alpha$-harmonic function $f$ satisfies

$$\lim_{r \to 1^-} f_r = f^* \in \mathcal{D}'(\mathbb{T}) \ \ (\alpha > -1),$$

then, for $z \in \mathbb{D}$, it can be expressed in terms of a *Poisson-type integral*

$$f(z) = \mathcal{P}_\alpha[f^*](z) = \frac{1}{2\pi} \int_0^{2\pi} P_\alpha(ze^{-i\theta}) f^*(e^{i\theta}) d\theta$$

where

$$\mathcal{P}_\alpha(z) = \frac{(1 - |z|^2)^{\alpha+1}}{(1 - z)(1 - \overline{z})^{\alpha+1}}$$

is the complex valued *$\alpha$-harmonic Poisson kernel* in $\mathbb{D}$. In the case $\alpha = 0$ we obtain the classical Poisson's kernel $\mathcal{P} \equiv \mathcal{P}_0$.

More generally, if $a, b \in \mathbb{R}$ are not negative integers and $a + b > -1$, the $(a, b)$-*Poisson kernel* and the $(a, b)$-*Poisson integral* for $f \in L^1(\mathbb{T})$ is defined by

$$K_{a,b}(z) := c_{a,b} \frac{(1 - |z|^2)^{a+b+1}}{(1 - z)^{a+1}(1 - \overline{z})^{b+1}}, \quad \text{where} \quad c_{a,b} = \frac{\Gamma(a + 1)\Gamma(b + 1)}{\Gamma(a + b + 1)}$$

and

$$\mathcal{K}_{a,b}[f](z) := \frac{1}{2\pi} \int_0^{2\pi} K_{a,b}(ze^{-i\theta}) f(e^{i\theta}) d\theta, \quad z \in \mathbb{D},$$

where $\Gamma$ is the Gamma-function. Clearly, the $(0, \alpha)$-Poisson integral is just the $\alpha$-Poisson integral.

## 2. Lipschitz Continuity of $\bar{\alpha}$-Harmonic Mappings

*2.1. An Introductory Result*

As a starting point of our investigation, we used Theorem 2 which can be found in Chen's paper [18]. This theorem gives some rather strong assumption on $g = -\overline{L}_\alpha u$ ($g \in C(\mathbb{D})$), as well as for the boundary values of $u$ (condition $(d)$ of Theorem 2), which are proven to be sufficient for Lipschitz continuity of $u$.

Before we formulate the basic result, we need to introduce some notions. We say that function $v : \mathbb{D} \to \Omega$, $v \in C^2(\mathbb{D})$ is a solution of the $\bar{\alpha}$-*Poisson equation* if we have that

$$\begin{cases} u(z) = f(z), & \text{if } z \in \mathbb{T}, \\ -(\overline{L}_\alpha)u(z) = g, & \text{if } z \in \mathbb{D}, \end{cases} \tag{4}$$

for some $g \in C(\overline{\mathbb{D}})$ in the sense that $u_r \to f \in L^1(\mathbb{D})$ when $r \to 1^-$, where $u_r(e^{i\theta}) = u(re^{i\theta})$. The family of such $u$ that are diffeomorphisms preserving an orientation of $\mathbb{D}$ will be denoted as $V_{\mathbb{D} \to \Omega}[g]$.

Behm [16] has found a solution to the Dirichlet problem for the $\bar{\alpha}-$Poisson's equation, for zero boundary values in the sense of distributions. In addition, Chen and Kalaj [17] derived a formula for general functions, which has prescribed arbitrary boundary values, using Olofsson's and Wittsten's [1] result. This is described in the following theorem.

**Theorem 1** ([17]). *Let a function $g \in C(\mathbb{D})$ satisfy the condition $(1 - |z|^2)^{\alpha+1} g \in L^1(\mathbb{D})$, where $\alpha > -1$ is arbitrary. If $u \in C^2(\mathbb{D})$ satisfies equation $-\overline{L_\alpha} u = g$ and if $u_r \to f \in L^1(\mathbb{T}), r \to 1^-$ where $u_r(t) = u(re^{it}), t \in [0, 2\pi)$, then*

$$u(\omega) = v(\omega) + G_\alpha[g](\omega) \text{ for every } \omega \in \mathbb{D},$$

*where*

$$v(\omega) = \frac{1}{2\pi} \int_0^{2\pi} \frac{(1-|\omega|^2)^{\alpha+1}}{(1 - e^{i\theta}\overline{\omega})(1 - e^{-i\theta}\omega)^{\alpha+1}} f(e^{i\theta}) \, \mathrm{d}\theta, \quad G_\alpha[g](\omega) = \iint_{\mathbb{D}} G_\alpha(z, \omega) g(z) \, \mathrm{d}x \, \mathrm{d}y, \quad (5)$$

*and $G_\alpha(z, \omega)$ denotes the Green function of the operator $\overline{L_\alpha}$, having the following form:*

$$G_\alpha(z, \omega) = \frac{(1 - \overline{z}\omega)^\alpha h(q(z, \omega))}{2\pi}, \text{ with } z \neq \omega,$$

$$h(r) = \frac{1}{2} \int_0^{1-r^2} \frac{t^\alpha}{1 - t} \, \mathrm{d}t, \ q(z, \omega) = \left| \frac{z - \omega}{1 - \overline{\omega}z} \right|.$$

In [18], Chen provided the following boundary characterizations of a Lipschitz-continuous $\overline{\alpha}$-harmonic mapping. Define $\underline{f}(t) = f(e^{it})$ and

$$S_\alpha[f](w) = \frac{1}{\pi} \int_0^{2\pi} \frac{(1 - |w|^2)^\alpha}{(1 - z\overline{w})^\alpha} \frac{\mathrm{Im}\,(w\overline{z})}{|z - w|^2} \underline{f}(t) dt, \tag{6}$$

where $z = e^{it}$. We remark that if $\alpha > 0$ and $\underline{f} \in L^\infty(\mathbb{T})$, then $S_\alpha[f]$ is bounded.

**Theorem 2** ([18]). *Let $g \in C(\overline{\mathbb{D}})$ and assume $u \in V_{\mathbb{D} \to \Omega}[g]$ has the representation*

$$u(\omega) = v(\omega) + G_\alpha[g](\omega),$$

*with $G_\alpha$ as in (5). If $\alpha \geqslant 0$, then the following four conditions are equivalent:*

*(i)*    *$u$ is (K, K')-qc and $\frac{\partial}{\partial r} v$ is a bounded function on $\mathbb{D}$.*
*(ii)*   *$u$ is Lipschitz on $\mathbb{D}$.*
*(iii)*   *$v$ is Lipschitz on $\mathbb{D}$.*
*(iv)*   *$f \in AC(\mathbb{T})$ is such that $f'$ belongs to the class $L^\infty(\mathbb{T})$ and $S_\alpha[\underline{f'}]$ is bounded on $\mathbb{D}$.*

In order to prove the main result of this paper, we need to show two refinements of the above result.

*2.2. Refinement of Part (iv) in Theorem 2*

Let $p \in (0, \infty]$. For a function $f : \mathbb{D} \to \mathbb{C}$ and $0 < r < 1$, we define

$$M_p(r, f) = \left( \frac{1}{2\pi} \int_0^{2\pi} |f(re^{i\theta})|^p \, d\theta \right)^{1/p}$$

and

$$\|f\|_p = \begin{cases} \sup_{0 < r < 1} M_p(r, f) & \text{for } p > 0, \\ \sup_{z \in \mathbb{D}} |f(z)| & \text{for } p = \infty. \end{cases}$$

The *generalized Hardy space* $\mathcal{H}_G^p(\mathbb{D})$ is the space of all measurable functions $f : \mathbb{D} \to \mathbb{C}$ for which $M_p(r, f)$ exists for each $0 < r < 1$ and $\|f\|_p < \infty$; see e.g., [19]. Moreover, *Hardy space* $\mathcal{H}^p(\mathbb{D})$ (resp. $h^p(\mathbb{D})$) is defined as the set of all analytic (resp. harmonic) functions in $\mathcal{H}_G^p(\mathbb{D})$ on $\mathbb{D}$.

**Definition 2** ([4]). *The Hilbert transformation $\mathcal{H}[\psi]$ of a function $\psi \in L^1([0, 2\pi])$ is given by*

$$\mathcal{H}[\psi](t) = -\frac{1}{2\pi} \int_{0_+}^{\pi} \frac{\psi(t+\varphi) - \psi(t-\varphi)}{\tan \frac{\varphi}{2}} \, d\varphi.$$

An alternative definition of the Hilbert transform is provided by the following property.

**Theorem 3** ([4]). *If $u = P[\psi]$ and $f = u + iv$ is analytic with $v(0) = 0$, then $\underline{v}(t) = \mathcal{H}[\psi](t)$ a.e., where we use the notation $\underline{u}(t) = u(e^{it})$.*

If $\psi \in AC([0, 2\pi])$ (consequently $\psi' \in L^1([0, 2\pi])$), it can be checked that

$$\frac{\partial u}{\partial \theta} = \mathcal{P}[\psi'].$$

Having in mind that the harmonic conjugate of $U = \frac{\partial u}{\partial \theta}$ is the function defined by $V = r\frac{\partial u}{\partial r}$, we know that

$$V(re^{i\theta}) = \mathcal{P}[\mathcal{H}(\psi')], \tag{7}$$

$$\left(\frac{\partial u}{\partial r}\right)(e^{i\theta}) = \mathcal{H}[\psi'](\theta) \quad \text{a.e.} \tag{8}$$

In Theorem 2 from [15], the authors used operator $\mathcal{S}$ (defined at (6)), which generalizes the Hilbert transform. In the case $\alpha = 0$, The radial limits of the operator $\mathcal{S}_0$ coincide with the Hilbert operator $\mathcal{H}$. The next claim shows that, in the case of $\alpha > 0$, this new operator is bounded on $L^\infty(\mathbb{T})$, which was not the case for the Hilbert transform.

**Claim 1.** *If $\alpha > 0$, then the operator*

$$\mathcal{S}_\alpha : L^\infty(\mathbb{T}) \to L^\infty(\mathbb{D})$$

*is bounded.*

**Proof.** Let $\alpha > 0$. Recall that

$$\mathcal{S}_\alpha[f](w) = \frac{1}{\pi} \int_0^{2\pi} \frac{(1 - |w|^2)^\alpha \operatorname{Im}(w\bar{z})}{(1 - z\bar{w})^\alpha |z - w|^2} \underline{f}(t) \, dt, \tag{9}$$

where $z = e^{it}$ and

$$\operatorname{Im}(w\bar{z}) = \frac{(w\bar{z} - 1) + (1 - \bar{w}z)}{2i}.$$

Hence,

$$|\operatorname{Im}(w\bar{z})| \leqslant |1 - \bar{w}z|.$$

In addition, as $z\bar{z} = 1$, we have

$$|z - w| = |1 - z\bar{w}|.$$

Using the previous inequalities, we obtain

$$|\mathcal{S}_\alpha[f](w)| \leqslant \frac{1}{\pi} \int_0^{2\pi} \frac{(1 - |w|^2)^\alpha}{|1 - z\bar{w}|^{\alpha+1}} dt |\underline{f}|_\infty \leqslant 2M_\alpha |\underline{f}|_\infty,$$

where $M_\alpha = \frac{\Gamma(\alpha)}{\Gamma(\frac{\alpha+1}{2})^2}$, see [1]. $\quad\square$

Remark in the case $\alpha = 0$,

$$\mathcal{S}_0[f](w) = \frac{1}{\pi} \int_0^{2\pi} \frac{\operatorname{Im}(w\bar{z})}{|z - w|^2} \underline{f}(t) \, dt = \tilde{\mathcal{P}}[f](w).$$

where $\tilde{\mathcal{P}}$ denotes the conjugate Poisson kernel,

$$\tilde{\mathcal{P}}(z) = \operatorname{Im} \frac{1 + z}{1 - z} = \frac{2\operatorname{Im} z}{|1 - z|^2}.$$

Moreover, if $f \in L^1(\mathbb{T})$, then $\tilde{\mathcal{P}}[f]$ has radial limits almost everywhere and there holds the relation

$$\lim_{r \to 1^-} \tilde{\mathcal{P}}[f](re^{i\theta}) = \mathcal{H}[f](e^{i\theta}).$$

If, in addition, we have $\mathcal{H}[f] \in L^1(\mathbb{T})$, then

$$\mathcal{S}_0[f] = \tilde{\mathcal{P}}[f] = \mathcal{P}[\mathcal{H}[f]].$$

**Theorem 4.** *If $\alpha > 0$, and $p \in (1, \infty]$ then the operator*

$$\mathcal{S}_\alpha : L^p(\mathbb{T}) \to \mathcal{H}_{\mathcal{G}}^p(\mathbb{D})$$

*is bounded.*

**Proof.** We will consider only the case $p \in (1, \infty)$. According to Jensen's inequality, we have

$$|\mathcal{S}_\alpha[f](w)|^p \leqslant \frac{1}{\pi} \int_0^{2\pi} \frac{(1 - |w|^2)^\alpha |\underline{f}(t)|^p}{|1 - we^{-it}|^{\alpha+1}} \, dt \left( \frac{1}{\pi} \int_0^{2\pi} \frac{(1 - |w|^2)^\alpha \, dt}{|1 - we^{-it}|^{\alpha+1}} \right)^{p-1}$$

Next, by Fubini's theorem, we obtain that

$$\frac{1}{2\pi} \int_0^{2\pi} |\mathcal{S}_\alpha[f](|w|e^{i\theta})|^p d\theta \leq \|f\|_{L^p}^p \left( \frac{1}{\pi} \int_0^{2\pi} \frac{(1 - |w|^2)^\alpha \, dt}{|1 - we^{-it}|^{\alpha+1}} \right)^p \leq (2M_\alpha)^p \|f\|_{L^p}^p.$$

Hence

$$\|\mathcal{S}_\alpha[f]\|_p \leq 2M_\alpha \|f\|_{L^p}.$$

$\square$

For the proof of the main result we also need a corollary of this theorem:

**Theorem 5** ([19]). *Let us assume that $u = \mathcal{P}_\alpha[\varphi]$ where $\varphi \in AC(\mathbb{T})$ is such that $\dot{\varphi} \in L^p(\mathbb{T})$ with $1 \leqslant p \leqslant \infty$, and $\alpha > -1$ is not equal to zero.*

(a)  *In the case $\alpha > 0$, we have that $\frac{\partial}{\partial \bar{z}} u, \frac{\partial}{\partial z} u \in \mathcal{H}_{\mathcal{G}}^p(\mathbb{D}) \subset L^p(\mathbb{D})$.*

(b)  *In the case $-1 < \alpha < 0$ and $p < -1/\alpha$, we get $\frac{\partial}{\partial \bar{z}} u, \frac{\partial}{\partial z} u \in L^p(\mathbb{D})$.*

(c)  *In the case $-1 < \alpha < 0$ and $p \geqslant -1/\alpha$, we can find a function $u$ that is $\alpha$-harmonic on $\mathbb{D}$ and satisfies the conditions $\frac{\partial}{\partial \bar{z}} u, \frac{\partial}{\partial z} u \notin L^p(\mathbb{D})$. In addition, we have that $\frac{\partial}{\partial \bar{z}} u, \frac{\partial}{\partial z} u \notin \mathcal{H}_{\mathcal{G}}^1(\mathbb{D})$.*

The following theorem is the first result of this paper and we use it in proof of our main result.

**Theorem 6.** *(a) Let $h$ be defined on $\mathbb{D}$ and assume that $h \in h^1(\mathbb{D})$. Then $h$ is Lipschitz if and only if $\underline{h}' \in L^\infty$ and $\mathcal{H}(\underline{h}') \in L^\infty$.*

*(b) Let $h$ be $\alpha$-harmonic on $\mathbb{D}$ for $\alpha > 0$ and assume that $h \in \mathcal{H}_{\mathcal{G}}^1(\mathbb{D})$. Then $h$ is Lipschitz if and only if $\underline{h}' \in L^\infty$.*

**Proof.** (a) For detailed proof of this part, see [4].

(b) We can prove this by using part *(a)* of Theorem 5 in the case $p = \infty$. $\square$

*2.3. K-Quasiconformal $(p,q)$-Harmonic Mappings and Hölder Continuity*

Let $z = x + iy \in G$ and let $u$ be a differentiable function in $z$ and let $df(z)$ denote the differential operator at the point $z$. Then we define

$$|u'(z)| = \max_{|h|=1} |df(z)h| = \left|\frac{\partial}{\partial z}u(z)\right| + \left|\frac{\partial}{\partial \overline{z}}u(z)\right|, \tag{10}$$

$$l(u'(z)) = \min_{|h|=1} |df(z)h| = \left|\left|\frac{\partial}{\partial z}u(z)\right| - \left|\frac{\partial}{\partial \overline{z}}u(z)\right|\right|,$$

$$J_u(z) = \left|\frac{\partial}{\partial z}u(z)\right|^2 - \left|\frac{\partial}{\partial \overline{z}}u(z)\right|^2.$$

We say that function $u_0 : [a,b] \to \mathbb{C}, -\infty < a < b < \infty$ is absolutely continuous on interval $[a,b]$, or shortly $u_0 \in AC([a,b])$, if for every $\epsilon > 0$ there is $\delta > 0$ such that whenever a finite sequence of pairwise disjoint sub-intervals $(a_k, b_k) \subset I$ satisfies $\sum_j (b_j - a_j) < \delta$, we have that

$$\sum_j |f(b_j) - f(a_j)| < \epsilon.$$

Recall that $ACL(G)$ denotes the class of functions that are absolutely continuous on the lines in domain $D$, i.e., the class of functions whose restriction to all intervals $I$ that are parallel to the coordinate axis belongs to the class $AC(I)$. A sense-preserving homeomorphism $u : G \to \Omega$ is $(K, K')$-quasiconformal (or shortly $(K, K')-$qc) if $u \in ACL(G)$ and there exist $K > 1$ and $K' > 0$, satisfying

$$|u'(z)|^2 \leqslant K|J_u(z)| + K', \tag{11}$$

for every $z \in G$. We say that $u$ is $K$–quasiconformal if it satisfies Formula (11) for $K' = 0$. For more information about quasiconformal mappings, see [4].

A. Khafallah and M. Mhamdi proved the following theorem, which can be seen as a improvement of part $(i)$ in Theorem 2.

**Theorem 7** ([14]). *Let $u = \mathcal{K}_{a,b}[f]$ be a K–quasiconformal, where $a + b \in (-1, \infty)$.*

- *If $f$ is $\beta$-Hölder continuous on $\mathbb{T}$ for $0 < \beta < 1$, then $u$ is $\beta$-Hölder continuous on $\mathbb{D}$.*
- *If $f$ is Lipschitz continuous on $\mathbb{T}$, then $u$ is Lipschitz continuous on $\mathbb{D}$.*

## 3. Lipschitz Continuity of $\bar{\alpha}-$Green Integral

In this subsection, we will prove that, instead of $g \in C(\overline{\mathbb{D}})$, we can use the assumption that $g \in C(\mathbb{D})$ can be such that $(1 - |z|^2)^\alpha g$ belongs to the class $L^\infty(\mathbb{D})$, in order to prove Lipschitz continuity of the $\overline{\alpha}$-Green integral $G_\alpha[g]$ of the function $g$. This fact will play an important part in the proof of our main result.

The following two estimates can be obtained by direct investigation of the Green function $G_\alpha$, and can be found in [18].

$$2\pi\left|\frac{\partial}{\partial \omega}G_\alpha(z, \omega)\right| \leqslant \alpha C_\alpha |1 - \overline{z}\omega|^{\alpha-1}\left(1 - \left|\frac{z-\omega}{1-\overline{\omega}z}\right|^2\right)^{\alpha+1}\left(1 - \log\left|\frac{z-\omega}{1-\overline{\omega}z}\right|^2\right)$$
$$+ \frac{(1-|z|^2)^{\alpha+1}(1-|\omega|^2)^\alpha}{2|1-z\overline{\omega}|^{\alpha+1}|z-\omega|}, \tag{12}$$

$$2\pi\left|\frac{\partial}{\partial \overline{\omega}}G_\alpha(z, \omega)\right| \leqslant \frac{(1-|z|^2)^{\alpha+1}(1-|\omega|^2)^\alpha}{2|1-\overline{z}\omega|^{\alpha+1}|z-w|}. \tag{13}$$

In order to start with our work, we will prove the following two technical lemmas.

**Lemma 1.** *If $\beta > 1$, then*

$$\int_0^{2\pi} \frac{dt}{|1 - r\rho e^{it}|^\beta} \preceq \frac{1}{|1 - r\rho|^{\beta-1}}$$

*for $0 < r, \rho < 1$.*

**Proof.**

$$\begin{aligned}
\int_0^{2\pi} \frac{dt}{|1 - r\rho e^{it}|^\beta} &= 2\int_0^\pi \frac{dt}{((1 - r\rho)^2 + 4r\rho \sin^2 \frac{t}{2})^{\beta/2}} \\
&\leqslant \int_0^\pi \frac{dt}{((1 - r\rho)^2 + c_1 t^2)^{\beta/2}} \leqslant |t = (1 - r\rho)u| \\
&\leqslant \int_0^{\pi/(1-r\rho)} \frac{(1 - r\rho)\, du}{(1 - r\rho)^\beta (1 + c_1 u^2)^{\beta/2}} \\
&\leqslant \frac{1}{(1 - r\rho)^{\beta-1}} \int_0^\infty \frac{du}{(1 + c_1 u^2)^{\beta/2}},
\end{aligned}$$

since the last integral converges, we have the desired result. □

**Lemma 2.** *There exists $c_2 > 0$ such that*

$$M_1(r) = \iint_{\mathbb{D}} \frac{dx\, dy}{|z - r|} \leqslant c_2$$

*for every $0 < r < 1$.*

**Proof.** Let us use the substitution $z - r = \rho e^{it}$, where $0 \leqslant t < 2\pi$, $0 < \rho < \rho(t) = |r - e^{it}| \leqslant r + 1$. Then

$$\iint_{\mathbb{D}} \frac{dx\, dy}{|z - r|} = \int_0^{2\pi} dt \int_0^{\rho(t)} \frac{\rho\, d\rho}{\rho} \leqslant \int_0^{2\pi} (r + 1)\, dt \leqslant 4\pi. \tag{14}$$

□

Let $|\omega| = r$,

$$I_1(\omega) = \iint_{\mathbb{D}} \frac{(1 - |z|^2)^{\alpha+1}(1 - |\omega|^2)^\alpha}{2|1 - \bar{z}\omega|^{\alpha+1}|z - \omega|}\, dx\, dy,$$

$$I_2(\omega) = \iint_{\mathbb{D}} |1 - \bar{z}\omega|^{\alpha-1} \left(1 - \left|\frac{z - \omega}{1 - \bar{\omega}z}\right|^2\right)^{\alpha+1} \left(1 - \log\left|\frac{z - \omega}{1 - \bar{\omega}z}\right|^2\right) dx\, dy.$$

Also, inequalities

$$|1 - \bar{\omega}\zeta| \geqslant 1 - |\omega| \quad \text{and} \quad |1 - \bar{\omega}\zeta| \geqslant 1 - |\zeta| \tag{15}$$

can easily be verified.

The following two lemmas are crucial for the main result of this section:

**Lemma 3.** *There exists $c_3 > 0$ such that*

$$I_1(\omega) \leqslant c_3(1 - |\omega|^2)^\alpha \tag{16}$$

*for every $|\omega| < 1$.*

**Proof.** Using (15), we get

$$I_1(\omega) \preceq (1 - |\omega|^2)^\alpha \iint_{\mathbb{D}} \frac{\mathrm{d}x\,\mathrm{d}y}{|z - \omega|}.$$

Since we can use the coordinate change $s = \frac{\omega}{|\omega|}z$, we can use Lemma 2 to get our result.  □

Let $\zeta = \varphi_\omega(z) = \frac{\omega - z}{1 - \overline{\omega}z}$. If $\omega \in \mathbb{D}$, we have that $\varphi_\omega$ is conformal automorphism of the unit disc $\mathbb{D}$. The following formulae can be easily checked:

$$z = \frac{\omega - \zeta}{1 - \overline{\omega}\zeta}, 1 - |z|^2 = \frac{(1 - |\zeta|^2)(1 - |\omega|^2)}{|1 - \overline{\omega}\zeta|^2}, \tag{17}$$
$$1 - \overline{\omega}z = \frac{1 - |\omega|^2}{1 - \overline{\omega}\zeta}, \mathrm{d}z = -\frac{1 - |\omega|^2}{(1 - \overline{\omega}\zeta)^2}\,\mathrm{d}\zeta$$

**Lemma 4.** *There exists $c_4 > 0$ such that*

$$I_2(\omega) \leqslant c_4(1 - |\omega|^2)^\alpha \tag{18}$$

*for every $|\omega| < 1$.*

**Proof.** By using the substitution $s = \frac{\omega}{|\omega|}\zeta$, and $s = \rho e^{it}$ we get

$$I_2(\omega) = I_2(r) = \iint_{\mathbb{D}} \frac{(1 - |\omega|^2)^{\alpha+1}(1 - |\zeta|^2)^{\alpha+1}}{|1 - \overline{\omega}\zeta|^{\alpha+3}}(1 - \log|\zeta|^2)\,\mathrm{d}\zeta\,\mathrm{d}\eta$$
$$= (1 - |\omega|^2)^\alpha \int_0^1 (1 - \rho^2)^{\alpha+1}(1 - r^2)(1 - \log\rho^2)\int_0^{2\pi} \frac{\mathrm{d}t}{|1 - r\rho e^{it}|^{\alpha+3}}\rho\,\mathrm{d}\rho.$$

Using Lemma 1, we get

$$I_2(r) \preceq (1 - |\omega|^2)^\alpha \int_0^1 \frac{(1 - \rho^2)^{\alpha+1}(1 - r^2)(1 - \log\rho^2)}{|1 - r\rho|^{\alpha+2}}\rho\,\mathrm{d}\rho.$$

Since $1 - r\rho \geqslant 1 - r$ and $1 - r\rho \geqslant 1 - \rho$, we have that

$$I_2(r) \preceq (1 - |\omega|^2)^\alpha \int_0^1 \rho(1 - \log\rho^2)\,\mathrm{d}\rho \leqslant c_4(1 - |\omega|^2)^\alpha$$

for some $c_4 > 0$, which does not depend on $0 \leqslant r < 1$.  □

We are now ready to formulate the main result of this section, which is the generalization of Lemma 3.4 in Chen's paper [18]. The proof of this result follows directly from Lemmas 3 and 4.

**Theorem 8.** *Let $g \in C(\mathbb{D})$ be such that $(1 - |z|^2)^\alpha g \in L^\infty(\mathbb{D})$ and let $\alpha > 0$ be arbitrary. Assume that $G_\alpha[g]$ is the $\overline{\alpha}$-Green potential of the function $g$, i.e.,*

$$G_\alpha[g](\omega) = \iint_{\mathbb{D}} G_\alpha(z, \omega)g(z)\,\mathrm{d}x\,\mathrm{d}y.$$

*Then $\frac{\partial}{\partial\omega}G_\alpha[g], \frac{\partial}{\partial\overline{\omega}}G_\alpha[g] \in L^\infty(\mathbb{D})$.*

**Proof.** By the assumption, we have that there exists $M > 0$ such that $|g(z)| \leqslant M(1 - |z|^2)^{-\alpha}$. Using (12) and (13), now we have that $\left| \frac{\partial}{\partial \omega} G_\alpha[g](\omega) \right| \leqslant M(\bar{I}_1(\omega) + \bar{I}_1(\omega))$ and $\left| \frac{\partial}{\partial \bar{\omega}} G_\alpha[g](\omega) \right| \leqslant M\bar{I}_1(\omega)$, where

$$\bar{I}_1(\omega) = \iint_{\mathbb{D}} \frac{(1 - |z|^2)(1 - |\omega|^2)^\alpha}{2|1 - \bar{z}\omega|^{\alpha+1}|z - \omega|} \, \mathrm{d}x \, \mathrm{d}y,$$

$$\bar{I}_2(\omega) = \iint_{\mathbb{D}} (1 - |z|^2)^{-\alpha} |1 - \bar{z}\omega|^{\alpha-1} \left( 1 - \left| \frac{z - \omega}{1 - \bar{\omega}z} \right|^2 \right)^{\alpha+1} \left( 1 - \log \left| \frac{z - \omega}{1 - \bar{\omega}z} \right|^2 \right) \mathrm{d}x \, \mathrm{d}y.$$

For estimating integral $\bar{I}_1$ we can use (15) and Lemma 2 to get

$$\bar{I}_1(\omega) \leqslant c_5 \iint_{\mathbb{D}} \frac{\mathrm{d}x \, \mathrm{d}y}{|z - \omega|} \leqslant 4\pi c_5.$$

After applying (17), we get that

$$\bar{I}_2(\omega) = \iint_{\mathbb{D}} \left( \frac{(1 - |\zeta|^2)(1 - |\omega|^2)}{|1 - \bar{\omega}\zeta|^2} \right)^{-\alpha} \frac{(1 - |\omega|^2)^{\alpha+1}(1 - |\zeta|^2)^{\alpha+1}}{|1 - \bar{\omega}\zeta|^{\alpha+3}} (1 - \log|\zeta|^2) \, \mathrm{d}\zeta \, \mathrm{d}\eta$$

$$= \iint_{\mathbb{D}} \frac{(1 - |\omega|^2)(1 - |\zeta|^2)|1 - \bar{\omega}\zeta|^\alpha}{|1 - \bar{\omega}\zeta|^3} (1 - \log|\zeta|^2) \, \mathrm{d}\zeta \, \mathrm{d}\eta$$

$$\leqslant 2^\alpha \int_0^1 (1 - \rho^2)(1 - r^2)(1 - \log\rho^2) \int_0^{2\pi} \frac{\mathrm{d}t}{|1 - r\rho e^{it}|^3} \rho \, \mathrm{d}\rho.$$

Again, from Lemma 1 and (15), we get that $\bar{I}_2$ is bounded on the unit disc $\mathbb{D}$, which gives our conclusion. $\square$

As a direct consequence of Theorem 8 and Theorem 6 we have the main result of this paper.

**Theorem 9.** *Assume that $g \in C(\mathbb{D})$ is such that $(1 - |z|^2)^\alpha g$ is bounded and suppose that $u(\omega) = v(\omega) + G_\alpha[g](\omega)$, where $v \in C(\overline{\mathbb{D}})$ is an $\bar{\alpha}$-harmonic function, for some $\alpha > 0$. If the boundary function $\underline{v}$ is Lipschitz, then $u$ is also Lipschitz continuous on $\mathbb{D}$.*

## 4. Discussion

The main result of this article is one possible version of Kellogg's theorem on a solution of the $\bar{\alpha}$-Poisson's equation with a prescribed boundary mapping, assuming that the boundary function has the Lipschitz continuity property. In the previous sections, we discussed novelties of our work. Here, we add further comments. As an original approach in this article, we mention using some elementary integral inequalities originating from the Hardy theory. This approach was used to prove the boundedness of the gradient, whereas some earlier papers used some complicated infinite summation methods instead. Our method leads to a result under weaker conditions on the $\bar{\alpha}$-Laplacian and leaves space for further improvement. For example, this approach can be used to prove similar results under the $\bar{\alpha}$-Laplacian-gradient condition, where certain other continuity properties of Riesz potentials can be used; see [7]. A similar method is thoroughly investigated in [21–24].

As usual, $\mathbb{R}^n = \{(x_1, x_2, \ldots, x_n) : x_1, \ldots, x_n \in \mathbb{R}\}$ and $|x| = \sqrt{x_1^2 + x_2^2 + \ldots + x_n^2}$ is the Euclidean norm of $x \in \mathbb{R}^n$. We recall that, for a differentiable function $f : G \to \mathbb{R}$ on a domain $G \subset \mathbb{R}^n$, its gradient vector $\nabla f$ and (assuming twice continuous differentiability) its standard Laplacian $\Delta f$ are

$$\nabla f = \left( \frac{\partial f}{\partial x_1}, \frac{\partial f}{\partial x_2}, \ldots, \frac{\partial f}{\partial x_n} \right) \quad \text{and} \quad \Delta f = \sum_{j=1}^n \frac{\partial^2 f}{\partial x_j^2}.$$

We say that function $f$ satisfies the Laplacian-gradient inequality on the domain $G$ if there exist positive constants $a, b$ such that

$$|\Delta f(x)| \leqslant a|\nabla f(x)|^2 + b \text{ for every } x \in G.$$

Let $G_1$ and $G_2$ be domains in $\mathbb{R}^n$ with $C^2$ boundaries. One of the results obtained in [7] says that, if every coordinate of a quasiconformal diffeomorphism $f : G_1 \to G_2$ satisfies the Laplace-gradient inequality, then $f$ is Lipschitz. The proof of this result is based on the Flattening the boundary method, with some use of continuity properties of Riesz potentials.

## 5. Concluding Remarks and Observations

It is interesting to mention one important application of our work. For positive integers $\alpha$, the Lipschitz continuity of $\alpha$-harmonic functions $f$ from the unit disc $\mathbb{D}$ onto a $C^2$-domain was proved in [15], where the harmonic extension of the boundary function $\underline{f}$ is $(K, K')$-quasiconformal. Lipschitzity of quasiconformal harmonic mappings between the unit ball $\mathbb{B}^n$ and a spatial domain with a $C^{1,\beta}$ boundary ($0 < \beta < 1$) was proved in [25]. At this point, for our next article, we can announce a result that generalizes two results that were previously mentioned, using the main result of this article. Namely, for any $\alpha > 0$, we expect to prove the Lipschitz continuity of $(K, K')$-quasiconformal solutions $f$ of the $\overline{\alpha}$-Poisson's equation that map $\mathbb{D}$ onto a $C^{1,\beta}$ domain, under the assumption that $\rho_\alpha^{-1}\overline{L}_\alpha f \in L^\infty(\mathbb{D})$.

**Author Contributions:** Conceptualization, M.M.; Investigation, M.M., N.M. and A.K.; Writing—review and editing, M.M., N.M. and A.K. All authors have read and agreed to the published version of the manuscript.

**Funding:** This research received no external funding.

**Data Availability Statement:** Not applicable.

**Acknowledgments:** The first author acknowledge the funding support received from Serbian Academy of Science and Arts, Belgrade; the third author acknowledge the funding support received from the Deanship of Research Oversight and Coordination (DROC) at King Fahd University of Petroleum and Minerals (KFUPM)

**Conflicts of Interest:** The authors declare no conflict of interest.

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
