# Peer review of "Lipschitz Continuity for Harmonic Functions and Solutions of the α¯-Poisson Equation"

_axioms, doi:10.3390/axioms12100998_

Round 1

Reviewer 1 Report

This paper "Lipschitz continuity for harmonic functions and solutions of the α¯ -Poisson equation" by  Miodrag Mateljevi´c , Nikola Mutavdži´c , Adel Khafallah

contains very interesting results.

The paper is well written, it does not contains mathematical errors.

I recommend its publication!

Author Response

Thank you very much for your recommendation. We attach file with the comments about revised version.

Author Response

Thank you for your very useful comments and suggestions. We attach cover letter with explanations of the revised article.

Reviewer 3 Report

Please find attached report.

Author Response

(The authors gave the same response as above.)

Round 2

Reviewer 2 Report

Line 55: it should be written "G1" instead of "G!".

Reviewer 3 Report

Please double check everything again.